# Group Layout Pattern and Outdoor Wind Environment of Enclosed Office Buildings in Hangzhou

**Xiaoyu Ying** [1], **Yanling Wang** [2,*], **Wenzhe Li** [2], **Ziqiao Liu** [2] and **Grace Ding** [3]

1 Department of Architecture, Zhejiang University City College, No. 51, Huzhou Street, Gongshu District, Hangzhou 310015, China; yingxiaoyu@zucc.edu.cn
2 Department of Architecture, Zhejiang University, No.866, Yuhangtang Road, Sandun Town, Xihu District, Hangzhou 310027, China; 21712123@zju.edu.cn (W.L.); 21812107@zju.edu.cn (Z.L.)
3 Department of Built Environment, University of Technology Sydney, Broadway, Ultimo NSW 2007, Australia; Grace.ding@uts.edu.au
\* Correspondence: 21712105@zju.edu.cn; Tel.: +86-1576-394-4820

**Abstract:** This paper presents a study of the effects of wind-induced airflow through the urban built layout pattern using statistical analysis. This study investigates the association between typically enclosed office building layout patterns and the wind environment. First of all, this study establishes an ideal site model of 200 m × 200 m and obtains four typical multi-story enclosed office building group layouts, namely the multi-yard parallel opening, the multi-yard returning shape opening, the overall courtyard parallel opening, and the overall courtyard returning shape opening. Then, the natural ventilation performance of different building morphologies is further evaluated via the computational fluid dynamics (CFD) simulation software Phoenics. This study compares wind speed distribution at an outdoor pedestrian height (1.5 m). Finally, the natural ventilation performance corresponding to the four layout forms is obtained, which showed that the outdoor wind environment of the multi-yard type is more comfortable than the overall courtyard type, and the degree of enclosure of the building group is related to the advantages and disadvantages of the outdoor wind environment. The quantitative relevance between building layout and wind environment is examined, according to which the results of an ameliorated layout proposal are presented and assessed by Phoenics. This research could provide a method to create a livable urban wind environment.

**Keywords:** CFD; enclosed building; wind environment; group layout; Hangzhou; China

## 1. Introduction

According to the statistics of the National Bureau of Statistics of China, as seen in Figure 1, the total population of the mainland increased from 1367.82 million to 1395.38 million during the five years from 2014 to 2018. From the perspective of urban and rural structures, the resident population of urban areas has increased from 749.16 million to 831.37 million. The proportion of the urban population to the total population (urbanization rate) increased from 54.77% to 59.58% [1], indicating that it is in the middle and late stages of urbanization. The urbanization process has led to a sharp increase in urban density and the scale of cities in China. Many environmental problems have become increasingly prominent. The increasingly rough urban underlying surface and highly concentrated anthropogenic heat emissions together form a special urban climate environment, mainly manifested as weak urban winds, the heat island effect, and impeded urban air circulation and pollutant diffusion [2]. Unreasonable architectural layouts or architectural forms create an outdoor static wind zone, which is not conducive to the spread of pollutants and exhaust gases during the spring and autumn, and which

is also not conducive to heat dissipation in the summer. These problems have promoted the public's vision of the wind environment in urban space [3].

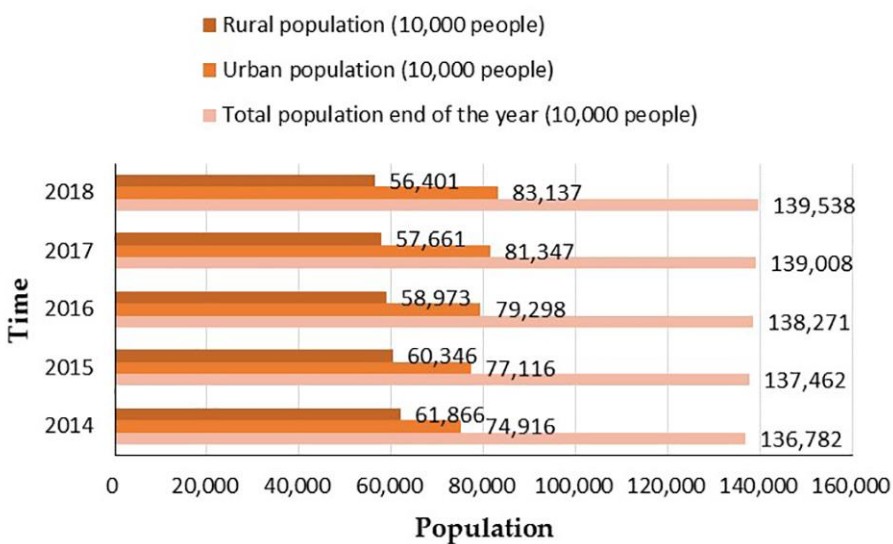

**Figure 1.** Mainland population statistics from China's National Bureau [1].

The office building is one of the most important types of architecture in the modern era. It accommodates a large number of working people in the city and has a profound impact on the urban spatial pattern. In addition to the place of residence, people spend the longest time in office spaces. With the development of society, people's demand for office buildings is also increasing. Office buildings are becoming more diverse, humane, low-carbon, and intelligent [4]. At present, research on office buildings focuses on form and function and lacks attention on factors affecting the outdoor environment.

The enclosed multi-story building layout can provide more natural lighting, create a better sense of territory and belonging, and reflect a traditional spatial mood. At the same time, because the enclosed space has good natural ventilation potential and is easy to create a relatively independent outdoor microclimate, it is increasingly used in urban modern architectural design [5]. The enclosed courtyard is often regarded as a microclimate modifier that improves the comfort conditions of the surrounding environment [6]. However, at present, the academic circles have rarely considered the natural ventilation performance of enclosed multi-story office buildings, and the association between typically enclosed office building layout patterns and the outdoor wind environment is still in the vague cognitive stage. In actual use, space utilization declines due to poor natural ventilation performance.

There are plenty of studies on wind motion through urban buildings [7]. Guo et al. [8] investigated the urban ventilation path, scattered morphology, and green space system that have a remarkable effect on promoting ventilation and alleviating the urban heat island effect. Enclosed city blocks are extremely unfavorable to ventilation. Kuo et al. explored the pedestrian-level wind flow characteristics inside the street canyon. The variables included the street canyon width, approaching wind direction, and podium height [9]. Guo et al. [10] used the art gallery as a case, and suggested that with comprehensive consideration given to elevation aesthetics and plane functions, one could create certain forms at suitable positions of a building to deflect wind and direct it through wind tunnels by distributing building volumes and creating open-up spaces and openings, so as to facilitate natural ventilation. Sharples et al. [11] carried out a wind tunnel study investigating the airflow through courtyard and atrium building models. The results from the study suggested that the small scale open courtyard in an urban environment had poor ventilation performance. Abdulbasit et al. [6] combined experimental and simulation methods in their research. The result verified that the manipulation of the courtyard configuration and its orientation impacted its microclimate modifying ability. Liu and Huang found that

based on the computational fluid dynamics (CFD) calculation and simulation results, the growth of the height of the patio has exerted the prime influence on the natural ventilation performance of the "Yinzi" dwellings, which were enclosed buildings [12]. Jin et al., focusing on severe cold regions, studied the relationships between the mean wind velocity ratio at the pedestrian level and the residential areas' building densities and high-rise buildings' layouts [13]. Xu et al. studied the traditional courtyards in southern Jiangsu Province. They proposed three strategies, including adjusting the courtyard layout, modifying the aspect ratio, and building an ecological buffer space to maximize its benefits of regulating the microclimate in winter and summer [14]. Ying et al. [15] analyzed the wind environment around a group of six square high-rise buildings. The research models in this study were six homogeneous cube models. Their study in 2019 [16] simulated the wind environment of 12 typical single-enclosed building opening schemes, as is shown in Figure 2 [16]. This research provided new ideas for the study of the outdoor wind environment of a single enclosed multi-story building. Overall, several investigations carried out simulations of the wind in enclosed buildings such as large-scale urban blocks, actual architecture cases, and homogeneous or single-volume models. There is no quantitative research on the outdoor wind environment of enclosed multi-story building groups in complex urban environments, which should be of great concern.

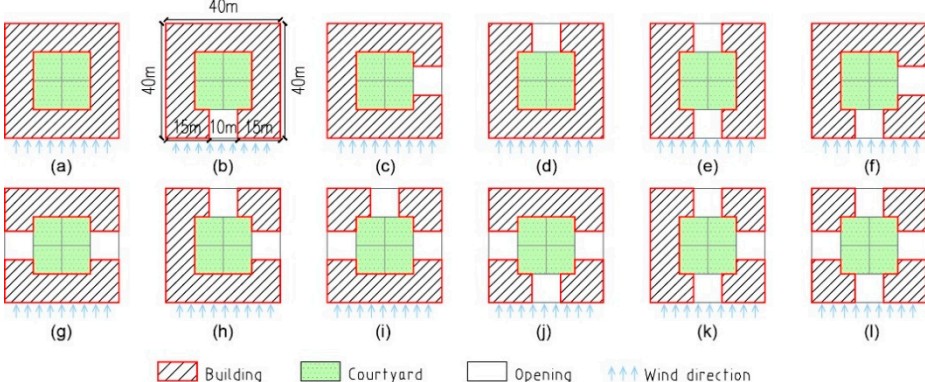

**Figure 2.** Twelve typical single-enclosed building opening schemes: (**a**–**l**), Opening scheme 1–12.

Therefore, it is necessary to research the effects of wind-induced airflows through the urban built layout pattern using statistical analysis and to investigate the association between typically enclosed office building layout patterns and the wind environment. This paper also proposes feasible improvement schemes for the unsatisfactory wind environment layout, providing a reference and design basis for architects. According to the possibility of architect design and the requirements of building fire protection [17], this article summarizes four typical enclosed office building group layouts. At the same time, for the convenience of comparative research, the model is simplified into four typical layouts: the multi-yard parallel opening, the multi-yard returning shape opening, the overall courtyard parallel opening, and the overall courtyard returning shape opening, which are explained in detail below.

At present, the research methods of the urban wind environment mainly include three methods: the base practical measurement method, wind tunnel experiments, and computer numerical simulation [18]. Computational numerical simulation is the main research method in current research. Due to the data analysis capabilities and experimental cost constraints, the base practical measurement method and wind tunnel experiments are relatively limited.

## 2. Materials and Methods

### 2.1. Methods

This paper obtained four typical multi-story enclosed office building group layouts according to Hangzhou City Planning Management Technical Regulations and the design specifications [17,19].

Then it researched the climate of Hangzhou and determined the comfortable range of the wind speed ratio, according to temperature and humidity, as the evaluation standard. Phoenics software was used to set the boundary conditions, simulate the natural ventilation performance of the four layout forms, and calculate the wind speed ratios of each measuring point. Finally, this paper used Excel software for data processing and comparative analysis. Moreover, the article also makes further research on layout forms with poor natural ventilation performance.

### 2.2. Boundary Conditions

The numerical simulation software Phoenics used in this paper based on Reynold's time-averaged equations can automatically select the required conditions for calculation. Boundary conditions were chosen accordingly.

#### 2.2.1. Inlet Wind Speed Distribution

The air movement is typically horizontal, showing less vertical behavior. However, in an urban environment, "topography" affects the wind motions. When the airflow passes through different areas and topographic zone (oceans, mountains, forests, cities, etc.), friction reduces the energy of the wind; when the wind speed is reduced, the wind structure (such as the turbulence degree, the swirl scale, etc.) can also change. The degree of change decreases with the increase in height, until at a certain height, the influence of ground roughness can be ignored. This layer of the atmosphere, which is affected by the friction of the earth's surface, is called the atmospheric boundary layer. The height of the atmospheric boundary layer varies with the meteorological conditions and the topographic and surface roughness. In general, the range of 300 m above the ground (not exceeding 1000 m) is within the scope of the atmospheric boundary layer, so that the wind speed above this range is not affected by the surface but can flow freely under the action of the atmosphere gradient.

#### 2.2.2. Mean Wind Speed Index Rate Distribution

In the atmospheric boundary layer, the average wind speed changes with altitude, and the variation is called wind shear or wind profile. At present, most countries use empirical exponential distribution to describe the change of average wind speed with altitude in near-surface layers. The velocity of approaching wind can be approximately described by a power-law profile [20] as follows:

$$U(z) = U_G \times (z/z_G)^\alpha. \tag{1}$$

In this formula $U(z)$ represents the average wind speed at any height $z$, $U_G$ is the average wind speed at the standard height $z_G$, and the index $\alpha$ is a parameter describing the ground roughness. The building in this article complies with category C in Table 1; thus $\alpha$ is 0.22. The gradient height $z_G$ was assumed to be 400 m.

**Table 1.** Standard four types of landforms in China [21].

| Category | Underlying Surface Properties | $\alpha$ | $z_G$/m |
|---|---|---|---|
| A | Offshore, island, sea, and desert areas | 0.12 | 300 |
| B | Fields, villages, jungles, hills, and towns and suburbs with sparse houses | 0.16 | 350 |
| C | Urban districts with dense buildings | 0.22 | 400 |
| D | Urban areas with dense buildings and high housing | 0.30 | 450 |

$\alpha$—Ground roughness coefficient, $z_G$—Gradient wind height in meters.

### 2.2.3. Boundary Conditions of All the Wall Surfaces

The research assumes that the airflow on the domain outlet has fully developed, and the airflow has returned to normal flow without building obstruction. Therefore, the outlet boundary is partially unidirectional. This paper used a no-slip wall boundary condition at all the wall surfaces and a normal zero gradient boundary condition at the domain outlet and the domain top as well as symmetrical boundary conditions at the two lateral boundaries of the domain. The boundary conditions for lateral and upper surfaces do not have significant influences on the calculated results around the target building because the computational domain is large enough [22–24].

### 2.3. Grid Size and Independence

Franke et al. [25] suggested using at least ten cells on each side of the building and at least three cells with pedestrian wind speeds at 1.5–2 m height above the ground. This paper used three kinds of grids (coarse, medium, and fine) at the central area including the coarse grid 15 m, the medium grid 7 m, and the fine grid 3 m in the X and Y axis. The grid independence study confirmed that numerical results with medium grids change little when the grids become finer. Thus, all other models used medium grids.

### 2.4. Domain Size

The setting of the calculation domain is related to the credibility of wind field simulation results. For the size of the computational domain, the blockage ratio should be below 3% based on knowledge of wind tunnel experiments [26]. Baetke et al. [27] defined the blockage ratio as the ratio of the frontal area of the cube to the vertical cross-sectional area of the computational domain. On the advice of Mochida et al. [22] and Shirasawa et al. [23], the lateral and the top boundary should be set 5 H or more away from the building, where H is the height of the target building. The outflow boundary should be set at least 10 H behind the building. Where the building surroundings are considered, the height of the computational domain should be set to correspond to the boundary layer height determined by the terrain category of the surroundings [20]. Following this suggestion, for this study, the domain size was 1200 m, 1200 m, and 400 m in the longitudinal (x), lateral (y), and vertical (z) directions, respectively.

### 2.5. Solution Methods and Convergence Condition

This paper chose the Realizable k-ε model to solve the problem. The governing equations used were those suggested by Cheng et al. [28]. Franke et al. [29] presented best practices guidelines for the CFD simulation of flows in the urban environment, developed within the COST Action 732 framework. Almost all the research articles related to this topic considered these guidelines as the best practice reference for urban wind CFD simulations. Initializing Reynolds-averaged Navier Stokes (RANS) simulations with uniform velocity, turbulent flow energy, and energy dissipation rate fields typically requires $10^3$ iterations to reach convergence. Calculation are conducted until the desired level of convergence is reached, i.e., the constant residuals of all equations are $10^{-4}$ or less [29,30]. The calculation conditions of the Phoenics are shown in Table 2.

**Table 2.** Computational condition of Phoenics.

| Computational Condition | Setting |
|---|---|
| Computational domain | $1200 \times 1200 \times 400$ m |
| Central meshing | Grid interval of 7 m in X and Y axis, 1 m in Z axis |
| Turbulence model | Standard k-ε turbulence model |
| Incoming flow speed | 2.7 m/s (summer), 3.8 m/s (winter), at the height of 10 m |
| Incoming flow direction | South (summer), north (winter) |
| Calculation rule | SIMPLEC model |
| Convergence condition | The maximum permissible residual error is $10^{-4}$ |
| Total number of iterations | 1000 |

### 2.6. Validation of the CFD Simulation

This article is a further study based on the wind environment of the enclosed multi-story building and the courtyard space. The authors validated the implemented CFD simulation method in previous research [15,16]. By using the same boundary conditions and solutions as in Zhang's study [31], the CFD simulation results were compared with wind tunnel experiments of similar buildings for experimental verification. It also showed that the CFD simulation results had good agreement with experimental results.

### 2.7. Layout Model Setting

According to the possibility of architect design and the requirements of building fire protection [17], this paper divided the layout of the enclosed office building group into the following two types, as seen in Figure 3: multi-yard type and overall courtyard type, including the multi-yard parallel opening (M-p), the multi-yard returning shape opening (M-r), the overall courtyard parallel opening (O-p), and the overall courtyard returning shape opening (O-r), as is shown in Figure 4. The ideal site model was set to 200 m × 200 m (length, width). The south side was the main road with a width of 28 m where the main entrance to the site was set up. The roads on the east, west, and north sides were secondary roads with widths of 21 m, and which had secondary entrances. The building group size was 79.5 m × 79.5 m × 24 m. The building was 8 m away from the road red line, the distance between the groups was 25 m, the width of the enclosed office building group was 15 m, the green space rate was not less than 30%, the building density was not more than 40%, the plot ratio was not more than 2.2, and the room depth was 18.4 m. These indicator parameters were selected according to Hangzhou City Planning Management Technical Regulations by the Hangzhou Planning Bureau [19]. The technical specifications of the four layouts shown in Table 3 met the above requirements.

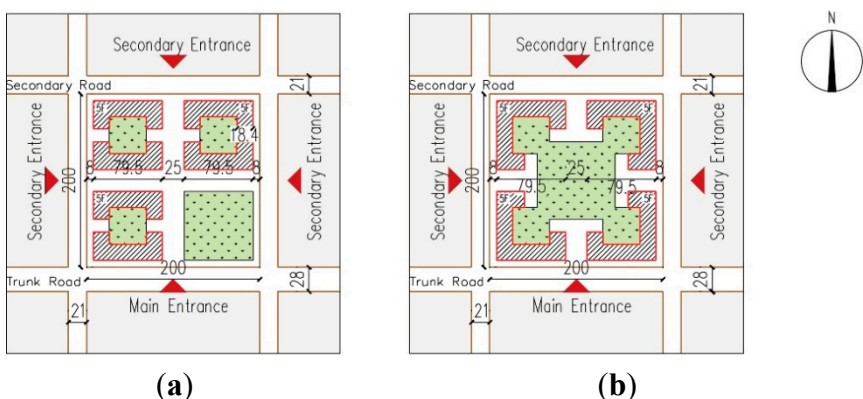

(**a**)                                            (**b**)

**Figure 3.** Two courtyard layout patterns. (**a**) Multi-yard type; (**b**) Overall courtyard type.

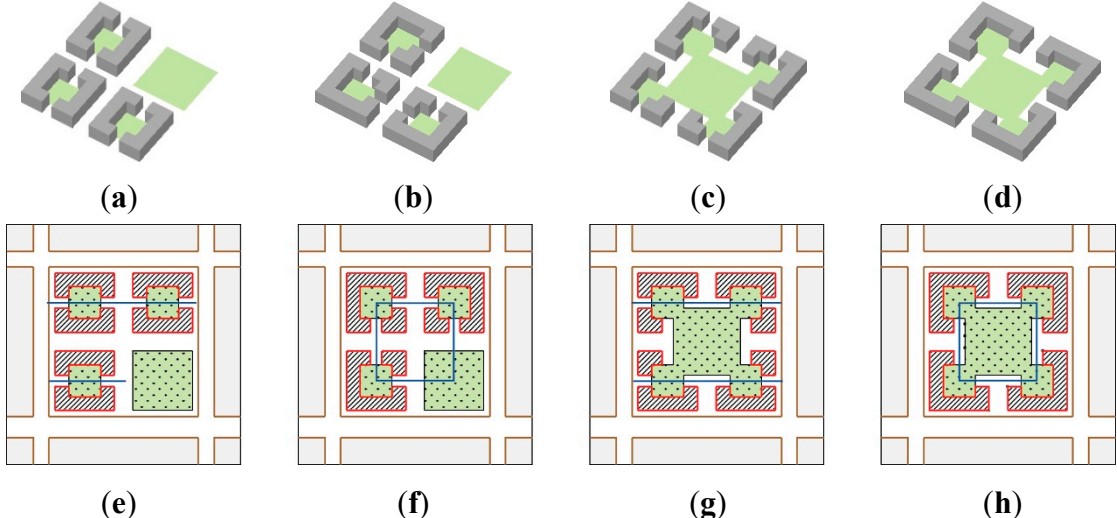

**Figure 4.** Four typical multi-enclosed office building group layouts. (**a**) M-p model; (**b**) M-r model; (**c**) O-p model; (**d**) O-r model; (**e**) M-p floor plan; (**f**) M-r floor plan; (**g**) O-p floor plan; (**h**) O-r floor plan.

**Table 3.** Technical specifications of four layouts.

| Index | M-p | M-r | O-p | O-r |
|---|---|---|---|---|
| Building density | 29.6% | 29.6% | 28.2% | 31.0% |
| Floor area ratio | 1.48 | 1.48 | 1.41 | 1.55 |
| Green space ratio | 29.5% | 29.5% | 36.3% | 36.3% |
| Building story number | 5 | 5 | 5 | 5 |
| Building height (m) | 24 | 24 | 24 | 24 |

*2.8. Evaluation Criterion*

This paper used Hangzhou as an example to have more practical significance. The research of the historical wind environment in Hangzhou indicated that the dominant wind direction in winter and summer in this region was obvious, and the wind frequency of the dominant wind direction was significantly higher than other wind directions. To simplify the calculation model, when simulating the wind environment around the building, the summer monsoon was set as the south wind (2.7 m/s) and the winter monsoon was set as the north wind (3.8 m/s) [32].

Many cities in the world have required evaluation of the wind environment before the construction of a building. The wind speed of the surrounding environment of the building was limited and required. Generally, the wind speed at a height of 1.5 m from the ground in the pedestrian zone is less than 5 m/s to meet the basic requirements that do not affect people's normal outdoor activities [33]. Stathopoulos et al. [34,35] suggested more data about a wider range of weather conditions and from different climates are needed to promote the new outdoor human comfort standards and described an approach towards the establishment of an overall comfort index taking into account, in addition to wind speed, the temperature and relative humidity in the urban area under consideration. The current evaluation methods mainly include relative trip comfort, wind speed probability statistics, and wind speed ratio evaluation methods. The relative trip comfort assessment method and wind speed probability statistical assessment method are both related to human subjective evaluation. The wind speed ratio is the ratio of wind velocity at each point (height = 1.5 m) to the wind velocity at the identical height at the inflow boundary, which reflects the degree of change in wind speed due to the presence of the building. The wind speed ratio equation is

$$R = V_0/V \tag{2}$$

where R is the wind speed ratio, $V_0$ is the velocity of a point, and V is the inflow velocity.

Kubota et al. [33] suggested that when the wind speed ratio is greater than 2.0, pedestrians will feel that the wind is too strong. On the other hand, people will not feel the presence of wind when the wind speed ratio is less than 0.5. Hyungkeun et al. found that pedestrians feel discomfort even at low wind speeds in winter. There are limits in assessing the comfort of pedestrians in winter since the existing criteria only consider the mechanical effects of the wind [36]. It is not reasonable to define pedestrian comfort value without considering other weather conditions, such as temperature and humidity. The average winter temperature of Hangzhou is −2.2 °C, the average relative humidity is 82% [32], and the climate comfort index should be greater than 25 according to the calculation method and grading principle of the climatic comfort index of the China State Meteorological Bureau [37]. Therefore, the winter wind speed should be less than 3.53 m/s, and the wind speed ratio less than 0.93 and more than 0.5. At the same time, the average summer temperature of Hangzhou is 32.4 °C, the average relative humidity is 62%, and the climate comfort index should be less than 80. Therefore, the summer wind speed should be greater than 1.25 m/s, and the wind speed ratio should be greater than 0.5 and less than 2.0.

## 3. Results and Discussion

Since wind is faster at the corner of buildings due to the less amount of topography, existing flow increases in speed at the corners in order to connect to the streamlines [38]. Therefore, the measurement points selected in the simulation were all locations with a large pedestrian flow and unfavorable wind speed, as is shown in Figure 5. Point $D_1$ was the measuring point in the courtyard where the summer monsoon may have adverse effects. Point $D_5$ was the measuring point in the courtyard where the winter monsoon may have adverse effects. $D_2$ and $D_6$ were measuring points with the concentrated pedestrian flow on the axis parallel to the wind direction and located at the corner of the building. $D_3$ and $D_4$ were the measuring points of concentrated human flow on the axis perpendicular to the wind direction.

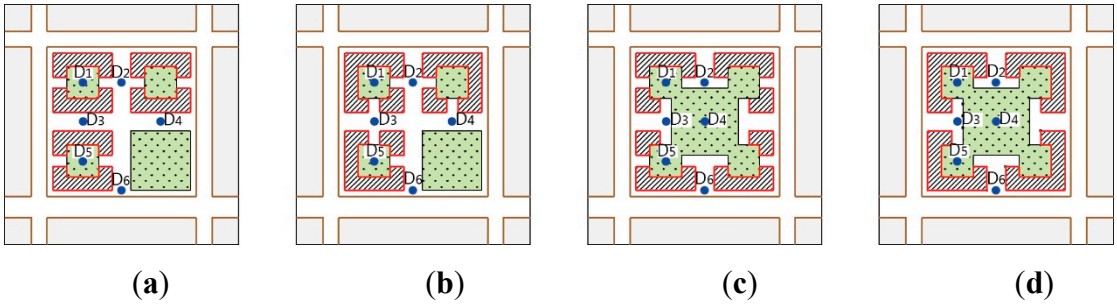

| (a) | (b) | (c) | (d) |

**Figure 5.** Distribution of the measuring points of four layouts. (**a**) Distribution of the measuring points of M-p; (**b**) Distribution of the measuring points of M-r; (**c**) Distribution of the measuring points of O-p; (**d**) Distribution of the measuring points of O-r.

### 3.1. Analysis of Wind Simulation Results in Summer

Figure 6 shows the simulation results of the wind environment at an outdoor pedestrian height (1.5 m) under the influence of the south wind in summer. The direction of the wind was south, and the wind speed was 2.7 m/s. The data of 6 measuring points of each scheme were statistically analyzed.

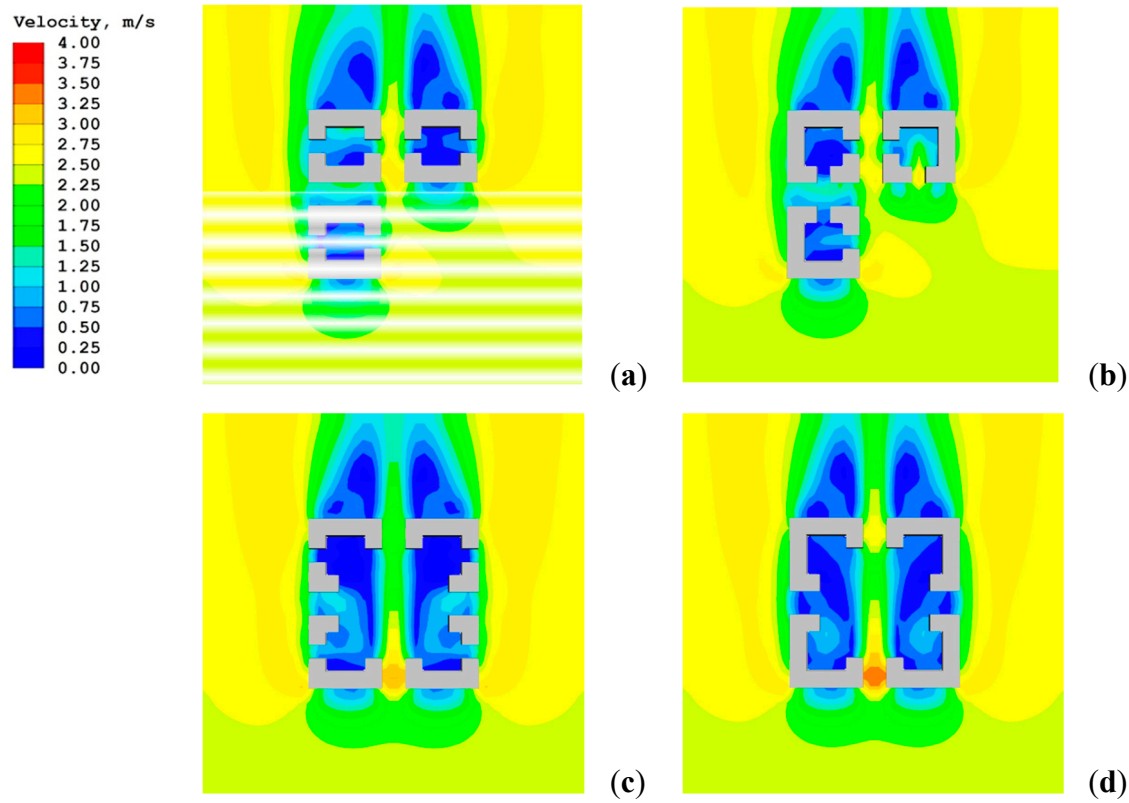

**Figure 6.** Summer simulation results of four layouts. (**a**) Summer wind simulation results of the M-p scheme; (**b**) Summer wind simulation results of the M-r scheme; (**c**) Summer wind simulation results of the O-p scheme; (**d**) Summer wind simulation results of the O-p scheme.

Figure 7 shows the contour map of the wind speed ratio at the outdoor pedestrian height (1.5 m) of each layout scheme.

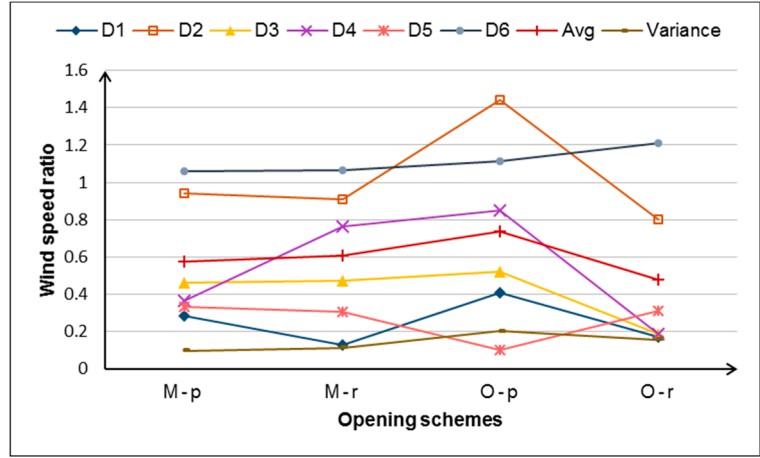

**Figure 7.** Measuring points' wind velocity ratio of summer wind in four schemes.

It can be seen intuitively from the figure that the wind speed of the measuring points on the axis parallel to the inflow wind direction was relatively large in the four layout schemes. They were the $D_2$ and $D_6$ measuring points in the multi-yard type and the $D_2$, $D_4$, and $D_6$ measuring points in the overall courtyard type. The second was the windward measurement point of the building, which was the multi-yard type measurement point $D_4$. The measured wind speeds in the courtyard and the measured wind speeds between the two buildings (north and south) were relatively small, namely the measuring

points $D_1$, $D_3$, and $D_5$. This also verified that it was easy to form tunnel wind on an axis parallel to the wind direction, so the wind speed at the measuring point on the axis was relatively large.

Besides, the wind speed ratio on the central axis consistent with the incident wind direction tended to decrease along the wind direction in general consideration. However, simulation results and data showed that only the O-p layout scheme met this situation, and other schemes were irregularly distributed, which indicates that the layout of the enclosed building group had a greater impact on the wind environment.

Among the four layout forms, the variance of measuring points of the O-p type was the largest, which was 0.20, and the wind speed ratio varied from 0.10 to 1.44. The variance of measuring points of the M-p type was the smallest, which was 0.10, and the wind speed ratio varied from 0.28 to 1.06. This indicated that the natural ventilation performance of each area of the O-p layout was very different, and pedestrians may feel uncomfortable. In Figure 8, the natural ventilation performance of the M-p layout was relatively uniform, and the wind speed ratio was closest to the comfort range (0.5–2.0), so the ventilation performance was excellent.

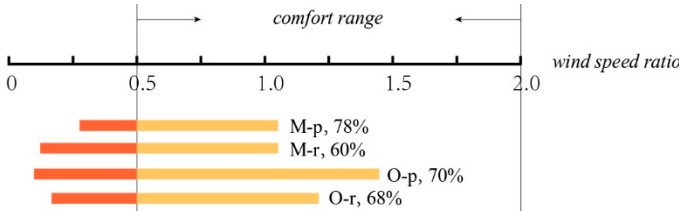

**Figure 8.** Comfort ratio of the four layouts (summer wind).

As can be seen from the Avg curve in Figure 7, the average wind speed ratio of the O-p type was the largest at 0.74, and the average wind speed ratio of the O-r type was the smallest at 0.48. The M-p and M-r types were 0.57 and 0.60, respectively. Therefore, the O-r layout was not conducive to outdoor ventilation because of the weak outdoor airflow. The evaluation of summer ventilation performance was as follows: M-p > M-r > O-p > O-r.

## 3.2. Analysis of Wind Simulation Results in Winter

Figure 9 shows the simulation results of the wind environment at an outdoor pedestrian height (1.5 m) under the influence of the north wind in winter. The direction of the wind was north, and the wind speed was 3.8 m/s. The data of 6 measuring points of each scheme were statistically analyzed.

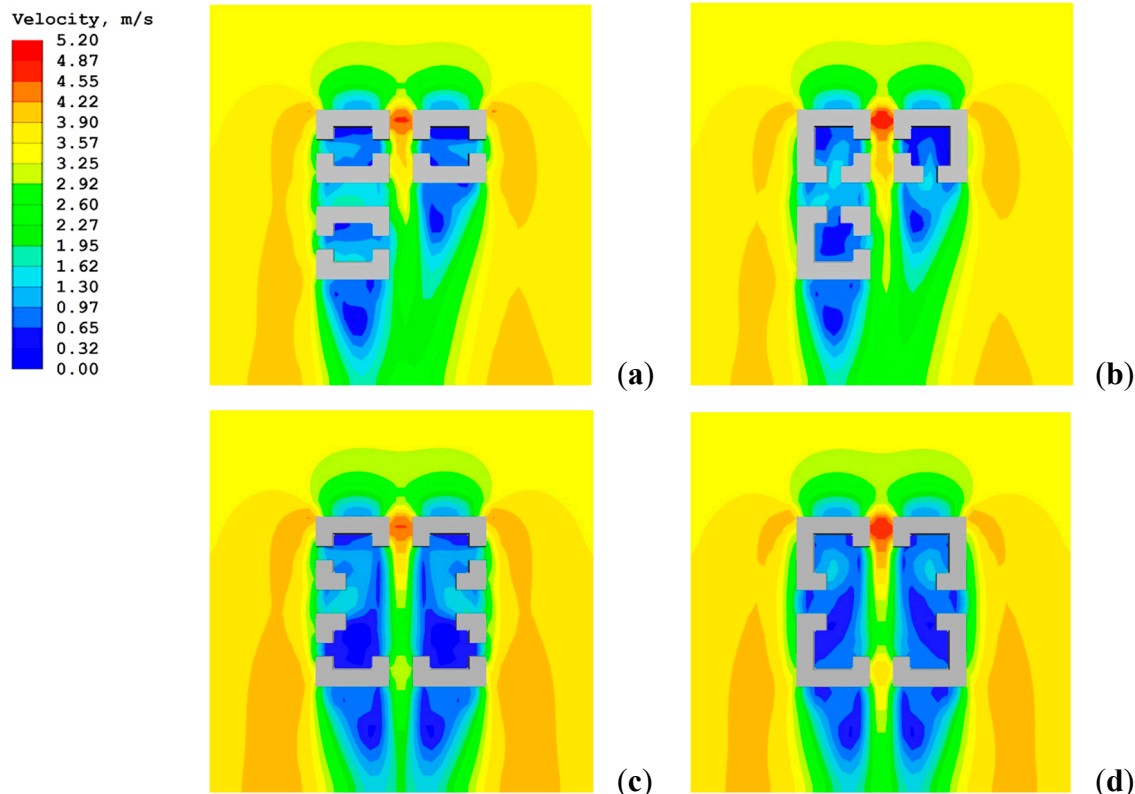

**Figure 9.** Winter simulation results of four layouts. (**a**) Winter wind simulation results of M-p scheme; (**b**) Winter wind simulation results of M-r scheme; (**c**) Winter wind simulation results of O-p scheme; (**d**) Winter wind simulation results of O-p scheme.

Figure 10 shows the contour map of the wind speed ratio at the outdoor pedestrian height (1.5 m) of each layout scheme.

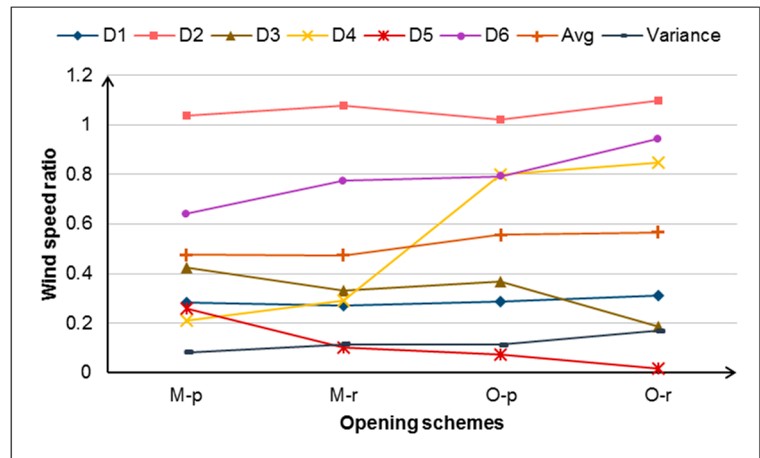

**Figure 10.** Measuring points' wind velocity ratio of winter wind in four schemes.

Similar to summer monsoon simulation results, the wind speed of the measuring points on the axis parallel to the inflow wind direction was relatively large in the four layout schemes. The actual wind speed exceeded 3 m/s in many places, so winter shelter measures should be considered. Among the four layout forms, the variance of measuring points of the O-r type was the largest, which was 0.17, and the wind speed ratio varied from 0.02 to 1.10. The variance of measuring points of the M-p type was the smallest, which was 0.08, and the wind speed ratio varied from 0.21 to 1.04. This indicates

that the natural ventilation performance of each area of the O-r layout was very different, and the natural ventilation performance of the M-p layout was relatively uniform, and the wind speed ratio was closest to the comfort range (0.5–0.93), as is shown in Figure 11.

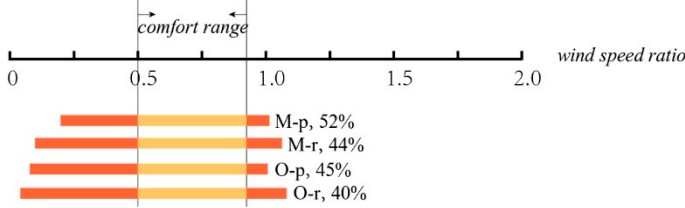

**Figure 11.** Comfort ratio of the four layouts (winter wind).

As can be seen from the Avg curve in Figure 10, the average wind speed ratio of the ratios of the four layout forms was between 0.47 and 0.57, which meets the comfort requirements of the winter monsoon. The evaluation of winter ventilation performance was as follows: M-p > M-r > O-p > O-r.

### 3.3. Influence of Overhead Ratio on Outdoor Wind Environment of Enclosed Building Group

From the analysis of the simulation results, results indicated that the outdoor wind environment of the O-r type was the most unfavorable. Therefore, taking the O-r type as an example, the influence of the overhead ratio on the outdoor wind environment of an enclosed building group was discussed. The model and measuring points were set, as shown in Figure 12 (the overhead ratio in this paper refers to the ratio of the ground floor area to the floor space of the building).

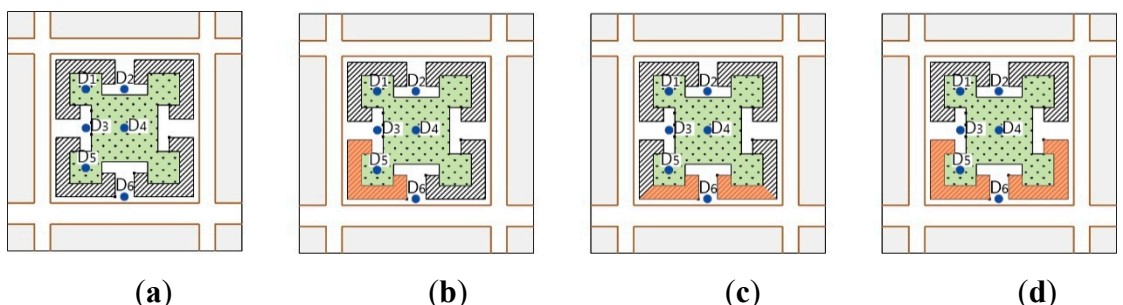

(**a**)          (**b**)          (**c**)          (**d**)

**Figure 12.** Distribution of the overhead measuring points. (**a**) Overhead ratio = 0; (**b**) Overhead ratio = 25% (one side); (**c**) Overhead ratio = 25% (middle); (**d**) Overhead ratio = 50%, the orange part represents the overhead position.

Figure 13 shows the outdoor wind environment under the influence of summer and winter winds when the ground floor of the southwest side was overhead. This indicates that when the ground floor was overhead, it had a great influence on the summer wind airflow and little influence on the winter wind airflow. It could meet the needs of summer ventilation and winter shelter that could create a comfortable outdoor wind environment.

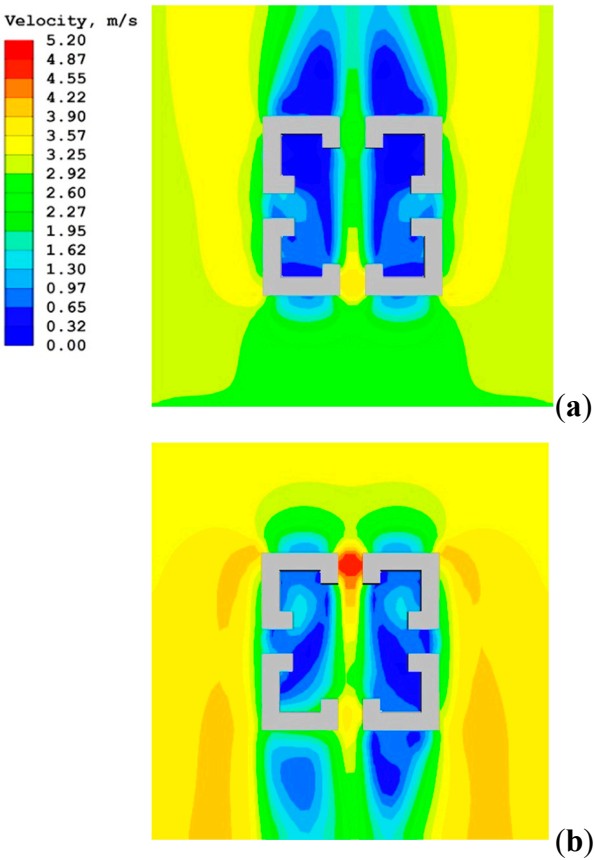

**Figure 13.** Simulation results with overhead layers using the overhead ratio of 25% on one side as an example. (**a**) Summer wind simulation result with overhead layers; (**b**) Winter wind simulation result with overhead layers.

Figure 14a is a contour map of wind speed ratios of four layout schemes under the influence of the summer monsoon. The overhead layer was on the side of the inflow direction, and the overhead ratio was 0, 25% (one side), 25% (middle), and 50%. The figure indicates that when the ground floor was overhead, it was more comfortable because of more airflow in summer. However, with the same overhead ratio, the outdoor wind environment was more comfortable when the overhead layer was in the middle (Figure 12c) than on one side (Figure 12b). As for the O-r layout, the simulation results of the overhead layers in the middle (Figure 12c) and the wind direction (Figure 12d) were very different, indicating that the wind conditions in the two cases were similar.

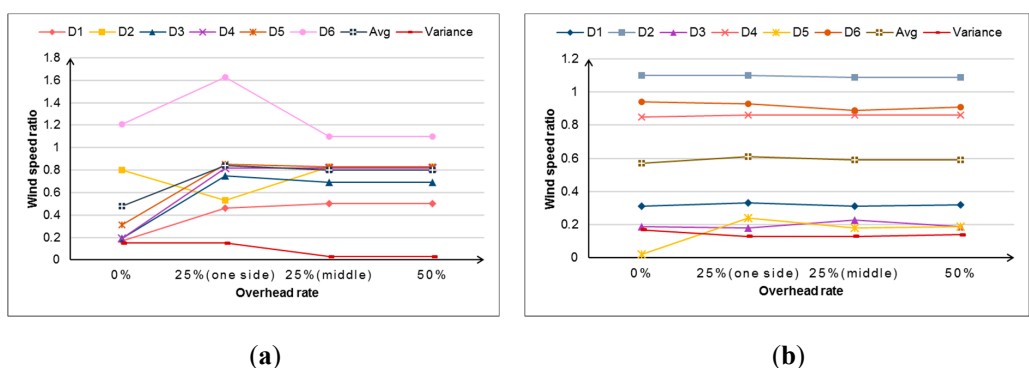

**Figure 14.** Wind velocity ratios when the overhead layers are in the inflow wind direction. (**a**) Wind velocity ratio of summer wind in O-r type; (**b**) Wind velocity ratio of winter wind in O-r type.

Figure 14b is a contour map of wind speed ratios of four layout schemes under the influence of winter monsoon. The overhead layer was on the side of the outlet direction, and the overhead ratio was 0, 25% (one side), 25% (middle), and 50%. This indicates that when the ground floor was overhead, the natural ventilation performance of the winter monsoon did not change much, and the effect of the overhead ratio on the outdoor wind environment was also small. This means that the impact of the overhead rate on the outdoor wind environment in winter can be ignored at this time.

When the overhead layer was located in the summer inflow direction, it could improve the problem of the unfavorable outdoor wind environment in the enclosed building group in the summer. However, it had little effect on the winter monsoon airflow. Therefore, setting up an overhead layer in the summer inflow direction is an effective measure to improve the outdoor wind environment of the O-r layout.

## 4. Conclusions

The enclosed space has good natural ventilation potential and application prospects, but the current actual situation is not very optimistic. Existing research lacks consideration of the natural ventilation performance of enclosed spaces and fails to solve the problem of low space utilization. Furthermore, the existing wind environment evaluation standards rarely consider the impact of temperature and humidity. The innovation of this article is to clarify the relationship between the layout of four typical enclosed office building group layouts (the M-p, M-r, O-p, O-r types) in Hangzhou and the comfort of the outdoor wind environment. At the same time, this article proposes improvement measures for the layout of poor natural ventilation performance. It studies the relationship between the overhead rate and the outdoor wind environment taking the O-r layout type as an example. Further research directions should clarify the parameter relationship between the two. Furthermore, the research method in this paper takes into account regional climate characteristics in more depth based on the best practice guidelines for CFD simulations. The conclusions of this paper can be summarized as follows:

The layout of the enclosed building group has a great impact on the outdoor wind environment, and the natural ventilation performance of the multi-yard type is better than the overall courtyard type. Among the four (M-p, M-r, O-p, O-r) layouts, the natural ventilation performance of the M-r layout type is the best, and the natural ventilation performance of the O-r layout type is the worst. The evaluation of natural ventilation performance is M-p > M-r > O-p > O-r.

In the enclosed building group it is easy to create a more comfortable outdoor wind environment in the courtyard space under the influence of the winter monsoon. However, appropriate measures should be taken where tunnel wind may be generated.

However, the courtyard space needs to consider measures to promote airflow in the courtyard space during the summer monsoon. It is an effective measure to use overhead layers locally. When the overhead ratio is 25% (middle), it not only has a high plot ratio but also can create a comfortable outdoor wind environment. The overhead ratio may have a certain functional relationship with the natural ventilation performance of the enclosed office building group, but more data is needed to verify this.

**Author Contributions:** X.Y. and Y.W. conceived the study, performed simulations and data analysis; X.Y. and Y.W. wrote the paper; W.L., Z.L. and G.D. provided constructive comments. All authors have read and agreed to the published version of the manuscript.

**Funding:** This research was funded by National Natural Science Foundation of China grant number 51878608; Natural Science Foundation of Zhejiang Province grant number LY18E080025; and Hangzhou Social Development self-declaration project grant number 20180533B08.

**Conflicts of Interest:** The authors declare no conflicts of interest.

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
