# Peer review of "Group Layout Pattern and Outdoor Wind Environment of Enclosed Office Buildings in Hangzhou"

_energies, doi:10.3390/en13020406_

Round 1

Reviewer 1 Report

GENERAL COMMENTS

The work is interesting, it has a certain degree of originality, and it might be appropriate for publication in the Energies journal, after performing a moderate level revision.

However, at this moment the proposed paper looks more like a technical report rather than a scientific paper and at the same time gives a general impression of negligence. Moreover, the work should be completed at various points.  From this perspective, the authors should work to improve the manuscript and prepare it better for a journal publication.

Some of the changes required are:

- most of the figures should be redesigned and some others should be slightly corrected;

- some sentences should be rephrased in order to express in a clearer way the ideas;

- the general clarity of the work should be improved;

- an additional check of the English grammar and spelling can be done;

- please check carefully the journal template. For example, you should write Figure X and not Fig. X. Please note that this is only an example and there are several similar issues that should be corrected.

Some specific comments are given next. They are not exhaustive, which means that there are still some other issues to be double-checked by the authors before resubmission.

SPECIFIC COMMENTS

ENGLISH LANGUAGE

This is in general OK. However, additional English grammar and spelling check should be performed. Moreover, some sentences have to be reformulated in order to present in a clearer way the ideas and the findings of the proposed work.

KEYWORDS

The second keyword is too long, please provide another. On the other hand, it would be suggestive to add the keywords  Hangzhou, China 

SYMBOLS AND EQUATIONS

There are only 2 equations provided. However, please provide a reference for equation (1).

ABBREVIATIONS

Please check carefully if all the abbreviations and notations considered in the work are explained for the first time when they are used, even if these are considered trivial by the authors. The paper should be accessible to a wide audience. For example, the meaning of PHOENICS should be explained.

FIGURES & TABLES

First, it is not clear why some figures are designed in black and white and some others in colours. Since this is an open-access journal it would be indicated to provide all figures in colours.

Furthermore, some corrections are required in the relationship with the figures and the figures captions, as follows:

Figures 2, 3, 4 – these figures should be designed in colours, you should denote each subplot as a), b),.. etc., and explain them in the figure caption;

Figure 5 – this figure should be designed in colours, you should denote each subplot as a), b),.. etc., and explain them in the figure caption; the measuring points should be more visible;

Figures 6, 9 –you should denote each subplot as a), b),.. etc., and explain them in the figure caption; the colorbar should be redesigned greater and the velocity values should be written with two decimals only;

Figures 7, 10 – these figures should be designed in colours;

Figures 8, 11 – these figures should be designed in colours, furthermore, it is not clear what units have the values provided on the horizontal axis;

Figure 12 –you should denote each subplot as a), b),.. etc., and explain them in the figure caption;

Figure 13 –you should denote each subplot as a), b),.. etc., and explain them in the figure caption; the colorbar should be redesigned greater and the velocity values should be written with two decimals only;

Figure 14 – this figure should be designed in colours.

PAPER STRUCTURE

In the introduction you should add more literature review, especially focusing on works published in MDPI journals.

Sections 3 and 4 are too short. You should join them in only one section, eventually, you may include them in section 2.

At Section 5 the first subtitle is written with different font.

In sections 5 or 6 you should better emphasise the originality and the added value brought by this work.

Author Response

Response

Dear reviewer,

Thank you for your patient review. Based on your suggestions, we have carefully revised the manuscript.

Some specific changes are given next.

ENGLISH LANGUAGE

We rechecked the grammar with Grammarly and removed some duplicates.

KEYWORDS

Already modified, added keyword Hangzhou, China.

SYMBOLS AND EQUATIONS

Added a reference for equation (1).

ABBREVIATIONS

Already modified, PHOENICS is a CFD simulation software.

FIGURES & TABLES

We changed the picture to color and modified figures and tables according to Energies template.

PAPER STRUCTURE

We added more literature review published in MDPI journals.

We included sections 3 and 4 in section 2.

The font of the article has been modified according to the template.

We have adjusted the structure of the article, added Method, emphasized the originality and added value of this work.

Reviewer 2 Report

The paper shows the results of CFD simulations regarding wind environement assessment of differenent office buildings configurations. The paper is well written and clear. The methodology suggested by the authors is robust, and the results are coherent.

Nevertheless, the paper has relevant drawbacks:

The paper is too much focused (including literature) on the Chinese framework. I would suggest to open a little bit the horizon to make the paper more interesting also for non Chinese readers. The novelty of the paper is unclear. Literature is reach of papers about CFD simulation of impact of different kind of buildings on the wind environment. I would suggest to stress more what is the novelty of the paper (i.e. the methodology applied?), if any. The presentation of the case study of Hangzhou city should be the final step of a more general methodology, which should be replicable in any location. Instead, the paper is focused only on the case study, which is very limiting. I would suggest to rewrite the paper by presenting the methodology designed by the authors, clearly divided by steps, so the same method can be applied by others. And the case study then can become an example on how to apply the methodology, being not any more the core of the paper.

Minor issues:

lines 34-41: repetition.

Figure 2: reference is missing.

Line 103: please justify why you think the temperature effects can be neglected.

Line 148: please give a definition of blockage ratio.

Line 161-162: please support the statement with some references.

Figures 6 and 9: please put below the figures the corresponding building acronym; and also avoid 6 zero after the comma, it has no sense.

Line 349: repetition "at the same time".

Author Response

Response

Dear reviewer,

Thank you for your patient review. We will fully consider your suggestions for further research. Based on your suggestions, we have carefully revised the manuscript. We have adjusted the structure of the article, added Method, emphasized the originality and added value of this work. Since we forgot to add line numbers to the previous version of the manuscript, it is not clear to us which lines you are referring to. But we did our best to correct all the issues.

Some specific changes are given next.

Lines 34-41: repetition.

Already modified.

Figure 2: reference is missing.

Added the reference to Figure 2.

Line 103: please justify why you think the temperature effects can be neglected.

This is an oversight and has been modified. This article considered the effects of temperature and humidity.

Line 148: please give a definition of blockage ratio.

Already added.

Line 161-162: please support the statement with some references.

We have deleted some redundant descriptions due to the adjustment of the structure and length of the article.

Figures 6 and 9

Already modified.

Line 349: repetition "at the same time".

Already modified.

Reviewer 3 Report

The correlation between space configuration (buildings/multi-yards/courtyards/layouts patterns) and the wind performance is somehow narrow due to its limited diversity concerning shapes/forms (what is built and what is in between the buildings). For instance, regarding the wind tunnel effect in the courtyard type, it could be interesting to analyze if a more generous space between building corners would bring changes towards the obtained results (or if different corners configurations) – this is to say that the simulation could have explored more situations with small alternative proposals regarding building/yard configurations so that more possibilities to compare would arise. If so, the extent of discussion/results would be richer. 

Author Response

Response

Dear reviewer,

Thank you for your patient review. We will fully consider your suggestions for further research. We have carefully revised the manuscript, adjusted the structure of the article, added Method, emphasized the originality and added value of this work.

Reviewer 4 Report

In the presented paper the quantitative relevance between building layout and wind environment is examined, according to the result of which an ameliorate layout proposal is presented and assessed by PHOENICS.  The manuscript would benefit from a broader treatment of the conclusions sections, including more comments justified by the results. The article was written carelessly and major revision is necessary.

The following comments are suggested to be addressed in the revised manuscript:

The paper have wrong reference numbers in the text and description of all items in References.

References in the lines 24-27 and in the title of Figure 1 are missing.

The same sentences in the lines 34-36 and 37-39.

Why is the drawing (Fig. 4) referred to in line 93 only put on page 6?

Where in the text is cited - Aly, A.-M., & Bresowar, J., 2016. Aerodynamic mitigation of wind-induced uplift forces on low-rise buildings: A comparative study. Journal of Building Engineering, 5:267-276 - from References?

What does "zG/m" mean in the Table 1?

From which References item the information given in the lines 152-154 is cited.

Are you sure that in the line 171 iterations is 10-4?

Incorrect Figure number in the line 178.

Incorrect Figure number in the line 181.

From which References item the information given in the lines 187-188 is cited?

Incorrect Table number on the page 6.

Table 2. (presented one the page 6) is not referred to in the text.

Figures 6 and 9 are not clear enough.

Figure 13 is not presented clearly. Figure 13 descriptions is missing.

Are you sure that in the line 317 should be Fig. 10?

Author Response

Response

Dear reviewer,

Thank you for your patient review. Based on your suggestions, we have carefully revised the manuscript, adjusted the structure of the article, added Method, emphasized the originality and added value of this work. Since we forgot to add line numbers to the previous version of the manuscript, it is not clear to us which lines you are referring to. But we did our best to correct all the issues.

Some specific changes are given next.

The format of the references has been modified. References in the lines 24-27 and in the title of Figure 1 are missing.

Already added.

The same sentences in the lines 34-36 and 37-39.

Already modified.

Why is the drawing (Fig. 4) referred to in line 93 only put on page 6?

 Already modified.

Where in the text is cited - Aly, A.-M., & Bresowar, J., 2016. Aerodynamic mitigation of wind-induced uplift forces on low-rise buildings: A comparative study. Journal of Building Engineering, 5:267-276 - from References?

There were references in the previous period, but they were not cited in the article. Already deleted.

What does "zG/m" mean in the Table 1?

zG- Gradient wind height in meters.

From which References item the information given in the lines 152-154 is cited.

We have deleted some redundant descriptions due to the adjustment of the structure and length of the article.

Are you sure that in the line 171 iterations is 10-4?

Already modified.

Incorrect Figure number in the line 178.

Incorrect Figure number in the line 181.

Already modified.

From which References item the information given in the lines 187-188 is cited?

Due to our previous negligence, we are not sure which lines you mean, but the contents of this section are fully quoted.

Incorrect Table number on the page 6.

Already modified.

Table 2. (presented one the page 6) is not referred to in the text.

Added, “The technical specifications of the four layouts in Table 3 meet the above requirements.”

Figures 6 and 9 are not clear enough.

Changed to clear pictures.

Figure 13 is not presented clearly. Figure 13 descriptions is missing.

Already modified.

Are you sure that in the line 317 should be Fig. 10?

Already modified.

Round 2

Reviewer 1 Report

Most of my observations have been considered and appropriate changes and corrections have been operated. There are however still some minor issues that have to be considered before publication.

Figure 1 - please write what is represented on the figure axes. Also, please redesign this figure, you should start on the y-axis with  the lowest year at the basis; Figure 2 - you should explain the subplots a-l in the figure caption; Figures 7, 10 - you should change the text on the x-axis to: opening schemes; Figure 13 is confusing because you use two different colormaps, please redesign the figures using the same colormaps for the two subplots and implicitly the same colorbar; Figure 14 - you should write on each subplot 'overhead rate' only once on the x-axes;

Author Response

Response

We have revised these detailed and important issues.

Figure 1

Already modified.

Figure 2

(a)– (l), Opening scheme 1–12.

Figures 7, 10

We have changed the text on the x-axis to: opening schemes.

Figure 13

Already modified.

Figure 14

We write on each subplot 'overhead rate' only once on the x-axes.

Reviewer 2 Report

The authors addressed most of the reviewers' comments. Nevertheless, the novelty of the paper is still not well highlighted in the text. The authors added some lines in the conclusions, but i) the lines have been put in the wrong place (they should be in the introduction), and ii) they are not convincing at all. As stated in my previous review report, there is a massive literature about outdoor wind environment, and the authors did not provide sufficient information to demonstrate the novelty/originality of their paper.

Author Response

We have revised some detailed and important issues. We made some changes to the conclusion section again. The research method of this paper is based on the existing wind environment research, according to the best practice guidelines of CFD simulation. It makes more in-depth considerations based on the regional climate characteristics in the wind environment assessment standards. This research method applies to any place, including Hangzhou, on the premise of knowing the climatic characteristics of regional temperature and humidity.

Reviewer 4 Report

The resubmitted paper has a better defined focus which is adequately developed and highlights potential research gaps to create a livable urban wind environment. The authors included the suggestions and observations in the original article. After revision the paper is ready for publication.

Author Response

We have revised some detailed and important issues. We made some changes to the conclusion section again. Thank you for your valuable suggestions.
